# HS-Surf: A Novel High-Frequency Surface Shell Radiance Field to Improve Large-Scale Scene Rendering

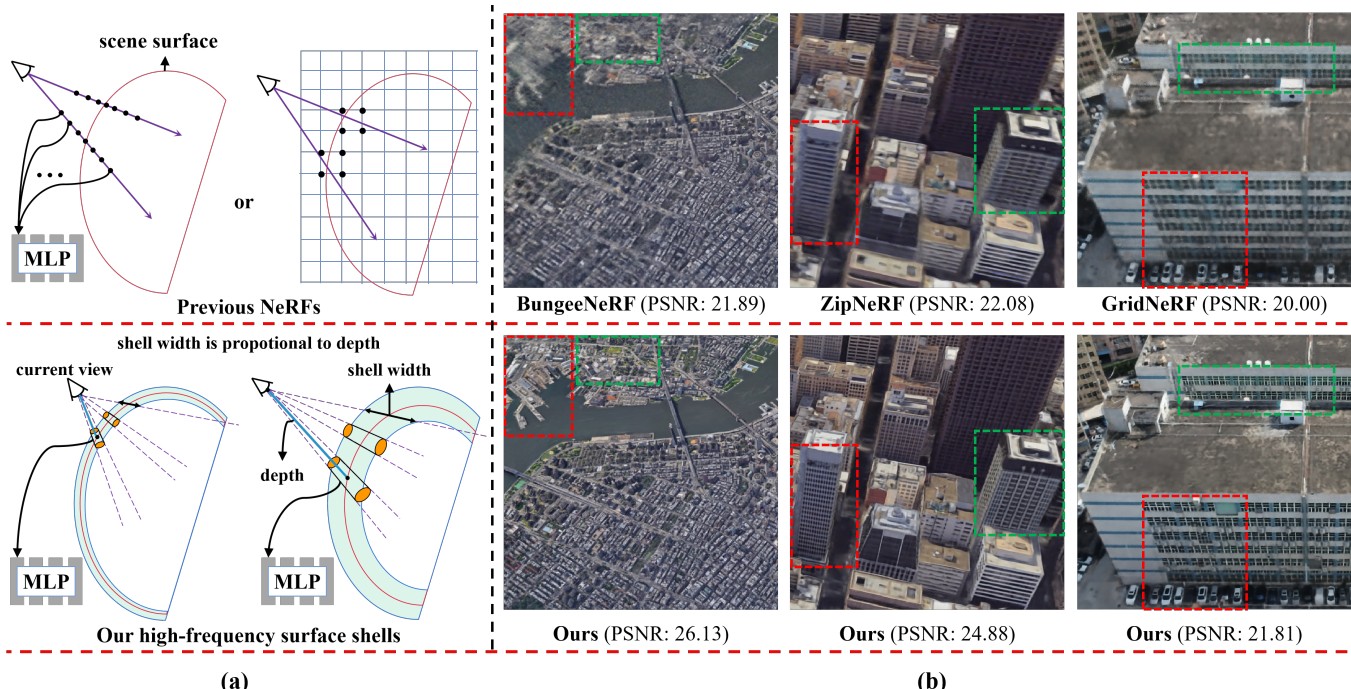

Figure 1: (a) Top: Prior NeRFs evenly sampling along the ray or on the grid, which usually falls in the contentless areas or is limited by the grid resolution. It leads to rendering blur and model capacity wasting. Bottom: HS-Surf constructs a shell on the scene surface based on the current view's depth, fully using model capacity on texture-rich areas to improve rendering quality. (b) Our method could render more high-frequency information on the scene surface to improve the clarity of textures compared to the SOTA NeRFs.

## ABSTRACT

Prior neural radiance fields often struggle to preserve high-frequency textures in urban and aerial large-scale scenes due to insufficient model capacity on the scene surface. This is attributed to their sampling locations or grid vertices falling in empty areas. Additionally, most models do not consider the drastic changes in distances. To address these issues, we propose a novel high-frequency surface shell radiance field, which uses depth-guided information to create a shell enveloping the scene surface under the current view, and then samples conic frustums on this shell to render high-frequency textures. Specifically, our method comprises three parts. Initially, we propose a strategy to fuse voxel grids and information of distance scales to generate a coarse scene at different distance scales. Subsequently, we construct a shell based on the depth information to carry out compensation to incorporate texture details not captured by voxels. Finally, the smooth and denoise post-processing further improves the rendering quality. Substantial scene experiments and ablation experiments demonstrate that our method achieves the obvious improvement of high-frequency textures at different distance scales and outperforms the state-of-the-art methods.

## CCS CONCEPTS

• **Computing methodologies → Image-based rendering**.

## KEYWORDS

large-scale scenes, high-frequency shell, surface rendering, high-frequency textures

*ACM MM, 2024, Melbourne, Australia*
© 2024 Copyright held by the owner/author(s). Publication rights licensed to ACM.
ACM ISBN 978-x-xxxx-xxxx-x/YY/MM
https://doi.org/10.1145/nnnnnnn.nnnnnnn

**Unpublished working draft. Not for distribution.**

# 1 INTRODUCTION

Rendering urban and aerial large-scale scenes has many applications like AR/VR and digital navigation. Prior neural radiance fields [1–3] (NeRFs) have tried improving the rendering quality, which can be categorized into two streams. The first [4–6] divides the scene or camera poses into multiple sub-regions or groups, and each unit is represented by a NeRF. This increases the number of NeRF modules, indirectly enhancing model capacity on the surface. However, those NeRFs sample along the entire ray, including empty spaces. The second [7, 8] reconstructs a coarse scene or density field to guide subsequent sampling, concentrating in high-density areas near the surface. Nevertheless, some samples inevitably fall into empty regions, such as the sampling interval ends.

Multi-layer perceptron (MLP) based NeRFs [1, 2] frequently sample along rays, resulting in many points falling into empty regions. Therefore, much model capacity for storing geometry and appearance is used to represent these meaningless spaces rather than the scene surface. Additionally, voxel [9] and grid-based [10–12] NeRFs only have a few vertices to be sampled near the target surface, leading to an upper bound on the model capacity allocated to the surface. These inefficient samplings result in a significant waste of model capacity, lacking enough model capacity on the scene surface to render high-frequency textures. Moreover, most NeRFs [4, 5, 7] have not considered the drastic changes in distances between the camera and the scene surface, which is prone to generate blurry rendering results at various distances.

To overcome the inefficient sampling and enhance the quality of high-frequency textures, we propose a novel high-frequency surface shell radiance field (HS-Surf) to efficiently increase model capacity on scene surfaces. It constructs a shell enveloping the scene surface based on the current view's scene depth. As shown in Figure 1(a), the shell's width increases with the depth, and conic frustums sampled on the shell are used to render high-frequency textures at different distance scales. We call this shell as High-Frequency Shell (HS). HS confines the rendering to scene surfaces, greatly enhancing the utilization of model capacity. Additionally, to model geometry and appearance at different distances, we propose a feature fusion strategy to embed conic frustums representing distances into voxel grids.

Our HS-Surf consists of three stages: initialization, compensation, and post-processing. The initialization uses hash-based voxel grids to generate coarse geometry and appearance. To model distances with drastic changes in large-scale scenes, the proposed feature fusion strategy embeds conic frustums into voxel grids. The compensation generates high-frequency textures at different distance scales. It first augments the coarse scene depth under the current view and constructs an HS based on the augmented depth. Conic frustums are then exclusively sampled on the shell to generate high-frequency textures lost in the coarse appearance. The post-processing uses a convolutional neural network (CNN) to smooth and denoise the rendering results to achieve a better visual effect.

The experimental results indicate that HS-Surf greatly improves high-frequency textures (see Figure 1(b)) and achieves state-of-the-art rendering quality. Additionally, we observe that our rendering

speed improves 2× to 4× faster than previous NeRFs, achieving double improvement of the rendering effect and computation efficiency. Our contributions can be summarized as follows:

- Our proposed high-frequency shell overcomes the sampling inefficiency of previous methods, efficiently increasing model capacity on scene surfaces to render high-frequency textures.
- The proposed feature fusion strategy embeds conic frustums into voxels to represent the distance scales, enabling the voxel to model the scene at various distances.

# 2 RELATED WORK

## 2.1 Neural Radiance Fields

NeRF [1, 3] employs MLPs to model volume density and color of spatial points. A lot of NeRF variants [13–23] render different size objects from small goods to large-scale scenes, aiming to enhance fidelity, rectify camera poses, and accelerate rendering. There are also models [24, 25] designed for unbounded scenes. To speed up rendering, some methods replace MLPs in NeRF with voxel [9, 26] or plane grids [10–12], but these increase GPU memory consumption. Recent methods [27–29] map the voxel vertices into smaller hash tables, which compress 3D spaces and achieve more compact representations.

MipNeRF [2] samples conic frustums along rays, and uses integrated positional encoding (IPE) of the frustums to represent the distance scales. However, the local continuous space of the frustum is incompatible with interpolation operations in grids. ZipNeRF [8] simulates the local space by sampling six points within a conic frustum. These points are then fed into InstantNGP [27]. 3D Gaussian [30] is another recent method with different mechanisms for scene representation, which involves fitting a large number of ellipsoids to approximate the target scene and render novel views.

## 2.2 Large-scale Scene Rendering

Some traditional methods [31–35] have been proposed to reconstruct the large-scale scenes. Their working pipeline usually needs three stages: keypoint detection, feature matching, and bundle adjustment. Keypoint detection [36–38] looks for unique and easily identifiable regions in images and constructs corresponding feature descriptions. Then, the features of key points are matched to compute camera poses and locations of 3D points. Finally, the camera poses and 3D points are jointly optimized by bundle adjustment [39, 40]. These methods can roughly reconstruct the target scene and synthesize novel views [41, 42], but the results often contain artifacts and holes.

The NeRFs-based methods are also introduced into large-scale scene rendering, including BungeeNeRF [6], BlockNeRF [4], MegaNeRF [5], URF [43], and GridNeRF [7]. BungeeNeRF divides camera poses into four groups based on their heights. A single NeRF is used in the highest group. For each subsequent group with lower heights, the model adds a NeRF module to capture the finer texture details. BlockNeRF and MegaNeRF divide the target scene into multiple sub-regions, with each region represented by a separate NeRF. By partitioning the camera poses or the scene, BungeeNeRF, BlockNeRF, and MegaNeRF reduce the target regions for each sub-NeRF, increasing the model capacity on scene surfaces. However, their

performance improvements are limited because NeRF still needs to sample the entire ray, including empty regions. The key issue of low utilization and allocation of model capacity on the scene surface remains unresolved. URF [43] leverages depth data of radar as auxiliary information to reconstruct street-level scenes.

GridNeRF [7] comprises a grid branch and a NeRF branch. The grid branch compresses the scene onto a ground plane to reconstruct a coarse radiance field, guiding a second sampling operation to add points in high-density regions near the scene surface. All the sampling points are then fed into the NeRF branch for rendering novel views. However, adding points near the surface still can have some samples falling into empty regions, particularly at the ends of the sampling interval. Moreover, the points in the first sampling operation are distributed across the entire ray, which increases the consumption of model capacity by empty regions. As a result, GridNeRF still lacks sufficient model capacity on the scene surface to render high-frequency texture details.

Both our HS-Surf and GridNeRF include rendering on the scene surface based on a coarse reconstruction. However, our method has different motivation and working mechanism. HS-Surf constructs high-frequency shells on the scene surface. These shells confine the computation of MLPs to the surface while excluding the surrounding empty regions, which greatly enhances the utilization of model capacity. Thus, our method has more power to render high-frequency textures.

## 3 METHOD

The overview of HS-Surf is illustrated in Figure 2, consisting of three stages: initialization, compensation, and post-processing. The initialization reconstructs the coarse geometry and appearance using hash-based voxel grids. IPE encoding of conic frustums and grid features are fused to model the target scene at different distance scales. The compensation first augments the coarse depth map under the current view to ensure a more accurate surface geometry. Subsequently, it constructs an HS based on the depth map, and samples conic frustums on HS to compensate for the lost high-frequency textures in the coarse appearance. This step is very important because the texture is attached to the geometry. Thus, good depth and its subsequent product of HS can confine sampling to effective texture areas. In the post-processing, a lightweight CNN is employed to smooth and denoise the rendering results of the compensation.

### 3.1 Initialization of Geometry and Appearance

Hash-based voxel grid is suitable for large-scale scenes as uniformly distributed vertices ensure that model capacity is reasonably allocated across the entire scene to generate a coarse radiance field. Additionally, hash tables are very useful for reducing GPU occupancy for high-resolution voxel grids.

Due to the drastic changes of distance scales in large-scale scenes, sampling points along rays can easily result in blurry rendering results at different distances. Inspired by MipNeRF [2], we sample conic frustums within the target scene and use IPE encoding to model distance scales. Specifically, a frustum is approximated by mean and covariance, which are then fed into IPE to generate the corresponding encoding [2]. Like MipNeRF, the radius of conic

containing the frustums at image plane $o + dir$ is set to $\dot{r}$, and $\dot{r}$ is the width of the pixel in world coordinates scaled by $2/\sqrt{12}$.

To leverage the advantages of both the hash-based voxel grids and conic frustums, we propose a feature fusion strategy to reconstruct the coarse scene at different distance scales. As shown in Figure 2, the center coordinates $x$ of the frustums are used to query features in the density grids $DG(\cdot)$ and the color grids $CG(\cdot)$. The IPE encoding $E_i$ of the frustums is fed into a two-layer MLP $M_i(\cdot)$, which is then fused with density features and color features to compute density $\sigma$ and color $rgb$ as follows:

$$\sigma = M_\sigma(concat(DG(x), M_i(E_i)))$$
$$rgb = M_c(concat(CG(x), M_i(E_i)), dir), \qquad (1)$$

where $M_\sigma(\cdot)$ and $M_c(\cdot)$ are small MLPs for generating density and color, and $dir$ is the ray direction. Volume rendering subsequently generates scene depth and rendering results.

The initialization stage can be described in two steps. The first step samples conic frustums along rays to render pixel colors $C_1$. The second step conducts finer sampling based on existing sample densities to obtain pixel depths $d_c$ and colors $C_2$. Both steps utilize the same voxel grids and hash tables. To ensure that the initialization results contain less artifacts such as "floaters" and "background collapse", we add an interval-based regularization loss $L_{dist}$ in the fine step, which is proposed in MipNeRF360 [24].

$$L_{dist}(s, w) = \sum_{i,j} \omega_i \omega_j \mid \frac{s_i + s_{i+1}}{2} - \frac{s_j + s_{j+1}}{2} \mid$$
$$+ \frac{1}{3} \sum_i \omega_i^2 (s_{i+1} - s_i), \qquad (2)$$

where $s$ and $w$ represent the (normalized) ray distances and weights of conic frustums in volume rendering, respectively. The role of $L_{dist}$ is to concentrate the frustums with high density into a narrower region. Then, the loss of the initialization stage is as follows:

$$L_{init} = \lambda_1 \parallel C_1 - C_{gt} \parallel_2^2 + \parallel C_2 - C_{gt} \parallel_2^2 + \lambda_2 L_{dist}, \qquad (3)$$

where $C_{gt}$ is the real pixel color. $\lambda_1$ and $\lambda_2$ are set to 0.1 and 0.001 in all experiments.

### 3.2 Compensation of Depth and Texture

The initialization includes a large number of voxel vertices far from the scene surface, leading to the insufficient model capacity on the surface and the loss of high-frequency textures. To recover the lost texture details, we construct a high-frequency shell for each view to efficiently increase the capacity on the surface. The details are as follows:

**Depth Augmentation.** The limited model capacity on the surface leads to coarse geometry with noises and holes. Therefore, before constructing the high-frequency shell, we propose a depth augmentation module to improve the depth map under the current view. As shown in Figures 2(a) and 2(c), the coarse depth $d_c$ is utilized to compute the coordinate $p_c$ of the scene surface. The depth augmentation then employs a four-layer MLP $D(\cdot)$ to predict the distance from $p_c$ to surface along the ray direction $dir$. The output of $D(\cdot)$ is added with $d_c$ to obtain a more accurate surface depth $d_f$ as follows:

$$d_f = D(\gamma(p_c), \gamma(dir)) + d_c, \qquad (4)$$

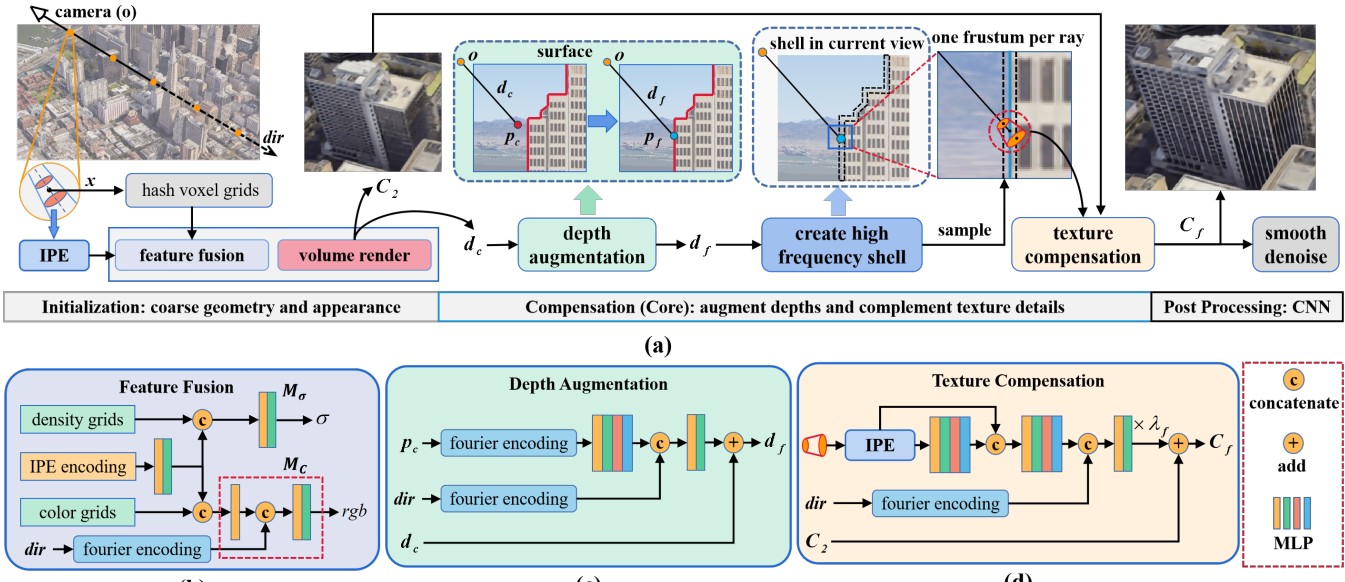

**Figure 2: Overview of HS-Surf. (a) is the method pipeline. The initialization employs the feature fusion to embed IPE encoding of conic frustum into the grid feature, generating the coarse geometry $d_c$ and texture $C_2$ at different distance scales. The compensation stage first augments the depth $d_c$ to obtain a more accurate depth $d_f$, and constructs a high-frequency shell of the current view based on $d_f$. Then, for each ray, a conic frustum sampled on the shell is fed into the texture compensation to render the high-frequency textures lost in $C_2$. Finally, the post-processing smooths and denoises the output of the compensation stage to achieve a better visual effect. (b), (c) and (d) are network structures for feature fusion, depth augmentation and texture compensation, respectively.**

where $\gamma$ represents the Fourier encoding [1]. More accurate depths and positions of the scene surface guarantee fewer errors in the inputs of subsequent modules.

We optimize the parameters of depth augmentation using both depth loss and subsequent rendering loss $L_{render}$. The depth loss ensures that the module preserves the basic geometry and structure of the scene, and the rendering loss $L_{render}$ is used to refine the scene depth for more accurate surface representation. Without depth ground truth, we take an augmentation manner to improve depth. Like the pyramid image processing, we take a down-sample on the coarse depth $d_c$ to get a low-resolution depth map, which could filter out certain noises while preserving the major depth information. The low-resolution depth map is determined by $W$ and is then compared with the output $d_f$ of depth augmentation to compute the depth loss $L_{depth}$. Specifically, we perform down-sampling in $d_c$ by a factor of 3:

$$W = (x\%3 == 0) \ and \ (y\%3 == 0)$$

$$L_{depth} = W \cdot \| d_f - d_c \|_2^2, \tag{5}$$

where $x$ and $y$ are the pixel coordinates.

**High-frequency Shell.** After obtaining accurate scene depth $d_f$ under the current view, we need to construct a high-frequency shell on the surface based on $d_f$. As shown in Figure 1(a), the shell's width $t_{range}$ along the ray determines the enclosed space on the ray $\{x|x = o + t \times dir, t \in [d_f - 0.5 \times t_{range}, d_f + 0.5 \times t_{range}]\}$. An infinite cone is constructed based on camera position, ray direction and pixel.

Then, we truncate the interval $[d_f - 0.5 \times t_{range}, d_f + 0.5 \times t_{range}]$ on conic axis to obtain a conic frustum $\Delta$.

As the depth increases, the high-frequency textures on the surface should become blurred or even disappear. Therefore, the width of $\Delta$ needs to increase with depth to suppress the high-frequency components in IPE encoding. The relationship between shell's width $t_{range}$ and depth $d_f$ follows a linear function. First, we calculate the normalized width $r$:

$$r = \lambda_r \cdot (k \cdot \frac{d_f - near}{far - near} + b), \tag{6}$$

where $\lambda_r = 0.1$ is a scaling factor to stabilize the model. *Near* and *far* are the distances traveled along the ray from camera to enter and exit the target scene. $k \geq 0$ and $b \geq 0$ are estimated by a four-layer MLP, and the input of MLP is the concatenation of *near* and *far*. The length $t_{range}$ of the shell in the world coordinate system is as follows:

$$t_{range} = max(min(r, \frac{1}{50}), \frac{1}{2000}) \cdot (far - near). \tag{7}$$

The normalized length $r$ needs to be clipped to $[1/2000, 1/50]$ for more stable results.

**Texture Compensation.** After sampling a conic frustum $\Delta$ for each ray on the high-frequency shell, we utilize continuous MLPs to complete the high-frequency textures lost in the voxel grids during initialization. As illustrated in Figures 2(a) and 2(d), the frustum $\Delta$ is located at the surface coordinate $p_f$, and its width equals to the shell's width along the ray. IPE encoding of $\Delta$ and ray direction

$dir$ are then fed into an eight-layer MLP $F(\cdot)$ to compute the high-frequency texture details missing in the voxel grids. The generated details are then added with the coarse rendering $C_2$ to obtain the complete appearance $C_f$ as follows:

$$C_f = \lambda_f \cdot F(IPE(\Delta), \gamma(dir)) + C_2, \tag{8}$$

where $\gamma$ is the Fourier encoding [1]. $\lambda_f = 0.2$ is a scale factor to stabilize the model. The output of $F(\cdot)$ is the texture residual instead of the complete rendering result. The reason is that this method can reduce the learning burden of MLPs and focus the attention of model on the generation of texture details.

Since the scope of inputs in the texture compensation is confined to the scene surface, the model capacity of MLP is dedicated to rendering textures on the target surface. This significantly improves both the capacity utilization on surface and the ability to render high-frequency details. The rendering loss $L_{render}$ is as follows:

$$L_{render} = \lambda_3 \parallel C_f - C_{gt} \parallel_1 + \parallel C_f - C_{gt} \parallel_2^2, \tag{9}$$

where $C_{gt}$ is the real pixel color, and $\lambda_3$ is set to 0.1. The loss function $L_{texture}$ of the compensation stage includes rendering loss $L_{render}$ and depth loss $L_{depth}$:

$$L_{texture} = L_{render} + L_{depth}. \tag{10}$$

## 3.3 Post-Processing of Smooth and Denoise

As the initialization and compensation stages calculate each pixel individually, the generated results may contain noises and are not continuously smooth. Therefore, a lightweight CNN-based post-processing is constructed to deal with them. The network contains two residual blocks to adjust the features of the original image $C_f$. Sigmoid activation is used in the final layer of convolution, which limits the output range to $[0, 1]$, whose details are provided in the supplementary material. We opt for a CNN due to the necessity of incorporating correlations between neighboring pixels in the smoothing and denoising processes. Convolution kernels offer a natural way to introduce such information. The loss function $L_{img}$ of the post-processing stage is as follows:

$$L_{img} = \lambda_4 \parallel C - C_{gt} \parallel_1 + \parallel C - C_{gt} \parallel_2^2, \tag{11}$$

where $C$ and $C_{gt}$ are the predicted and real colors, and $\lambda_4 = 0.1$.

## 3.4 Scene Division and Details

The compensation stage can further enhance the quality of high-frequency textures by partitioning the target scene. Assuming the scene is uniformly divided into $N$ sub-regions, and each contains a depth augmentation and a texture compensation, with corresponding MLPs represented as $D_i(\cdot)$ and $F_i(\cdot)$. The outputs of all sub-regions are combined into the final results. Therefore, the output $d_f$ of depth augmentation in Equation (4) is modified as follows:

$$d_f = \frac{\sum_{i=1}^{N} M_i \cdot (D_i(\gamma(p_c), \gamma(dir)) + d_c)}{\sum_{i=1}^{N} M_i}, \tag{12}$$

where $M_i = 1$ indicates that $p_c$ is located in the $i$-th region, otherwise $M_i = 0$. The output $C_f$ of texture compensation in Equation (8) is modified as follows:

$$C_f = \frac{\sum_{i=1}^{N} M_i \cdot (\lambda_f \cdot F_i(IPE(\Delta), \gamma(dir)) + C_2)}{\sum_{i=1}^{N} M_i}, \tag{13}$$

where $M_i = 1$ indicates that $p_f$ is located in the $i$-th region, otherwise $M_i = 0$. The increment in the number of MLPs leads to an augmentation in model capacity on the scene surface.

In Figure 2, the depth $d_c$ generated by the initialization needs to go through a gradient stop. Otherwise, the depth augmentation performance may decrease. In the initialization, the first coarse and second fine step samples 64 and 128 inters along a ray, respectively. The minimum resolution of the voxel grids is $256^3$, and the maximum resolution is $8192^3$ after 15 increments. The size of hash table is $2^{21} \times 4$ or $2^{22} \times 4$, and the hidden nodes of MLPs are 128. For the compensation stage, the hidden nodes of MLPs in the depth augmentation, estimation of high-frequency shell, and texture compensation are set to 256, 64, and 512. The channel of CNN in the post-processing is set to 32.

The training of HS-Surf consists of two stages. The first stage involves joint training of the initialization and compensation. The second stage only trains the post-processing. Their losses are as follows:

$$L_{stage1} = L_{init} + L_{texture} \tag{14}$$

$$L_{stage2} = L_{img}. \tag{15}$$

The learning rate is $1e - 4$ for both stages and decays exponentially to $1e - 5$ during training. More details can be found in the supplementary material.

# 4 EXPERIMENTS AND RESULTS

## 4.1 Experiment Setup

Our experiments are conducted on six scenes, including $Transamerica$, $56Leonard$, $Building$, $rubble$, $residence$, and $campus$. All models are implemented in environments of Python and PyTorch on a single RTX 3090 24G GPU. $Transamerica$ and $56Leonard$ are two synthetic scenes from the satellite level to the ground level, provided in BungeeNeRF [6]. All images are collected from Google Earth Studio [44], where the camera rotates around the central object of the scene, and the distance to the ground gradually decreases. $Building$, $rubble$, $residence$, and $campus$ are four real aerial data. The first two are from Mill 19 [5], and the remaining two are from UrbanScene3D [45]. In these four datasets, the drone always keeps a stable flying height and shoots to the ground along parallel lines or grid tracks. Therefore, the data distribution is uniform.

HS-Surf is compared with the previous state-of-the-art NeRFs, including BungeeNeRF [6], MegaNeRF [5], and GridNeRF [7]. Mip-NeRF [2] and ZipNeRF [8] are also used in the experiments because they are basic neural rendering methods and can model different distance scales in large-scale scenes. Since each image in the four aerial photography datasets has a different exposure and white balance, we refer to NeRF-in-the-wild [46] to assign a 48-dimensional appearance embedding for each image to model the lighting information. The highest frequencies of positional and directional encodings are set to 12 and 4 for all models. The details of training and testing sets, and more model configurations can be found in the supplementary material.

HS-Surf is also compared to 3D Gaussian [30] with different mechanisms, and the experimental results are presented in the supplementary material. The ability of 3D Gaussian to render high-frequency textures is not as good as HS-Surf because ellipsoids may

**Table 1: Performance comparison of HS-Surf with previous NeRFs on large-scale scenes**

| Model | Transamerica | | | | 56 Leonard | | | | Building | | | |
|---|---|---|---|---|---|---|---|---|---|---|---|---|
| | PSNR ↑ | SSIM ↑ | LPIPS ↓ | Time (s) | PSNR ↑ | SSIM ↑ | LPIPS ↓ | Time (s) | PSNR ↑ | SSIM ↑ | LPIPS ↓ | Time (s) |
| MipNeRF [2] | 22.12 | 0.6016 | 0.4856 | 50.90 | 21.87 | 0.5754 | 0.4883 | 51.48 | 19.44 | 0.3853 | 0.6499 | 44.32 |
| ZipNeRF [8] | 23.34 | 0.7092 | 0.4327 | 38.73 | 24.39 | 0.7864 | 0.3376 | 39.31 | 20.47 | 0.5282 | 0.5010 | 36.74 |
| BungeeNeRF [6] | 22.40 | 0.6216 | 0.4812 | 92.16 | 22.15 | 0.6015 | 0.4839 | 93.12 | × | × | × | × |
| MegaNeRF [5] | × | × | × | × | × | × | × | × | 20.69 | 0.4738 | 0.5544 | 251.96 |
| GridNeRF [7] | 23.22 | 0.6769 | 0.4640 | 89.30 | 23.47 | 0.6875 | 0.4605 | 90.41 | 21.00 | 0.5055 | 0.5259 | 80.85 |
| HS-Surf | **25.59** | **0.8304** | **0.2941** | **25.42** | **26.41** | **0.8679** | **0.2363** | **24.51** | **21.88** | **0.6039** | **0.4417** | **27.35** |
| Model | Rubble | | | | Residence | | | | Campus | | | |
| | PSNR ↑ | SSIM ↑ | LPIPS ↓ | Time (s) | PSNR ↑ | SSIM ↑ | LPIPS ↓ | Time (s) | PSNR ↑ | SSIM ↑ | LPIPS ↓ | Time (s) |
| MipNeRF [2] | 22.12 | 0.3933 | 0.6761 | 44.20 | 20.21 | 0.4504 | 0.6582 | 55.66 | 20.89 | 0.3687 | 0.7631 | 55.19 |
| ZipNeRF [8] | 23.68 | 0.5536 | 0.5169 | 37.15 | 21.00 | 0.5424 | 0.5240 | 44.77 | 20.61 | 0.4013 | 0.6591 | 48.58 |
| BungeeNeRF [6] | × | × | × | × | × | × | × | × | × | × | × | × |
| MegaNeRF [5] | 23.10 | 0.4591 | 0.6003 | 232.34 | 20.45 | 0.4869 | 0.5796 | 341.34 | 21.71 | 0.4028 | 0.6981 | 271.03 |
| GridNeRF [7] | 23.20 | 0.4752 | 0.5897 | 81.11 | 20.85 | 0.4967 | 0.5883 | 103.38 | 20.00 | 0.3863 | 0.6596 | 100.32 |
| HS-Surf | **24.24** | **0.5824** | **0.4943** | **27.01** | **22.12** | **0.5982** | **0.5015** | **34.06** | **21.97** | **0.4639** | **0.6183** | **34.36** |

be difficult to split into small enough sizes in large-scale scenes. Additionally, due to the lack of continuity in spherical harmonics, 3D Gaussian generates lots of aliasing under the unseen views in the training set.

## 4.2 Experiment results

In Table 1, we use PSNR, SSIM, LPIPS (VGG), and the time of rendering a frame to compare the rendering performance. The data distribution in *transamerica* and 56*Leonard* [6] is uneven because the cameras make loop shoot for the centers of scenes. Therefore, *transamerica* and 56*Leonard* are not divided. MegaNeRF [5] is not suitable for this mode, so it has no corresponding results. For the four aerial photography data, MegaNeRF and HS-Surf divide each scene into 8 sub-regions evenly. Since the drone always keeps a stable flight height, BungeeNeRF [6] cannot divide the camera poses according to the height from the camera to the ground, so it has no corresponding results on the later four datasets.

In *transamerica* and 56*Leonard* [6], the distance scales undergo drastic changes, HS-Surf achieves noticeable improvements in all metrics compared to models designed for variable distance scales (MipNeRF [2], ZipNeRF [8] and BungeeNeRF [6]), as shown in Table 1. The LPIPS errors of our method are reduced by 30%-40%. Figure 3 demonstrates a visual comparison example in the *transamerica* from the satellite level to the ground level. HS-Surf renders more high-frequency texture details for objects with different distances and shapes, which benefits from that HS-Surf embeds conic frustums into voxels to model distances and uses compensation to render high-frequency details at different distance scales.

In aerial photography scenes (*building*, *rubble*, *residence*, *campus*) with stable heights, HS-Surf also demonstrates better performance, with LPIPS error decreasing by 10%-20% compared to MegaNeRF [5] and GridNeRF [7]. Figure 4 presents visual comparison examples in *building* (top) and *residence* (bottom), HS-Surf synthesizes the most accurate views, especially regarding high-frequency textures for buildings with complex structures, wall or roof planes with different directions, and the vehicles with tiny height difference.

**Table 2: Comparison of model sizes**

| model | parameters | model | parameters |
|---|---|---|---|
| MipNeRF | 0.608M | ZipNeRF | 671.839M |
| BungeeNeRF | 1.080M | HS-Surf (w/o comp & pp) | 537.039M |
| MegaNeRF | 11.322M | HS-Surf (w/o pp) | 555.611M |
| GridNeRF | 465.914M | HS-Surf | 555.678M |

Compared to the other models, these cases validate the advantage of our high-frequency shells in confining the rendering to the surface and improving the utilization of model capacity. GridNeRF reconstructs a coarse scene using ground plane grids, then guides the sampling of NeRF branch in the high-density regions rather than on the surface. Its shortcomings are similar to MipNeRF, BungeeNeRF, and MegaNeRF. The inefficiency of sampling results in low capacity utilization and loss of high-frequency textures, as shown in Figure 4.

We find that the rendering of HS-Surf is 2× to 4× faster than other NeRFs in Table 1. Except for benefiting from the voxel grids, there are two other contributing reasons: 1) The compensation stage only samples a conic frustum for each ray. Thus, each ray only needs to be calculated once. 2) The hidden channel of shallow CNN in the post-processing is only 32. MipNeRF, BungeeNeRF, MegaNeRF and GridNeRF query multiple samples along a ray in each sampling stage, resulting in multiple computations for rendering a pixel. ZipNeRF samples six points in a conic frustum, bringing it a huge cost and a slow speed.

The model sizes on the dataset of *building* are presented in Table 2, where the result of BungeeNeRF are from *transamerica*. The compensation and post-processing are denoted as *comp* and *pp*. The compensation stage of HS-Surf introduces extra parameters on the scene surface, resulting in a parameter increase of 3.5%.

## 4.3 Ablation Study

In Table 3, we present an ablation study of HS-Surf. The post-processing stage is represented by PP. We summarize the findings

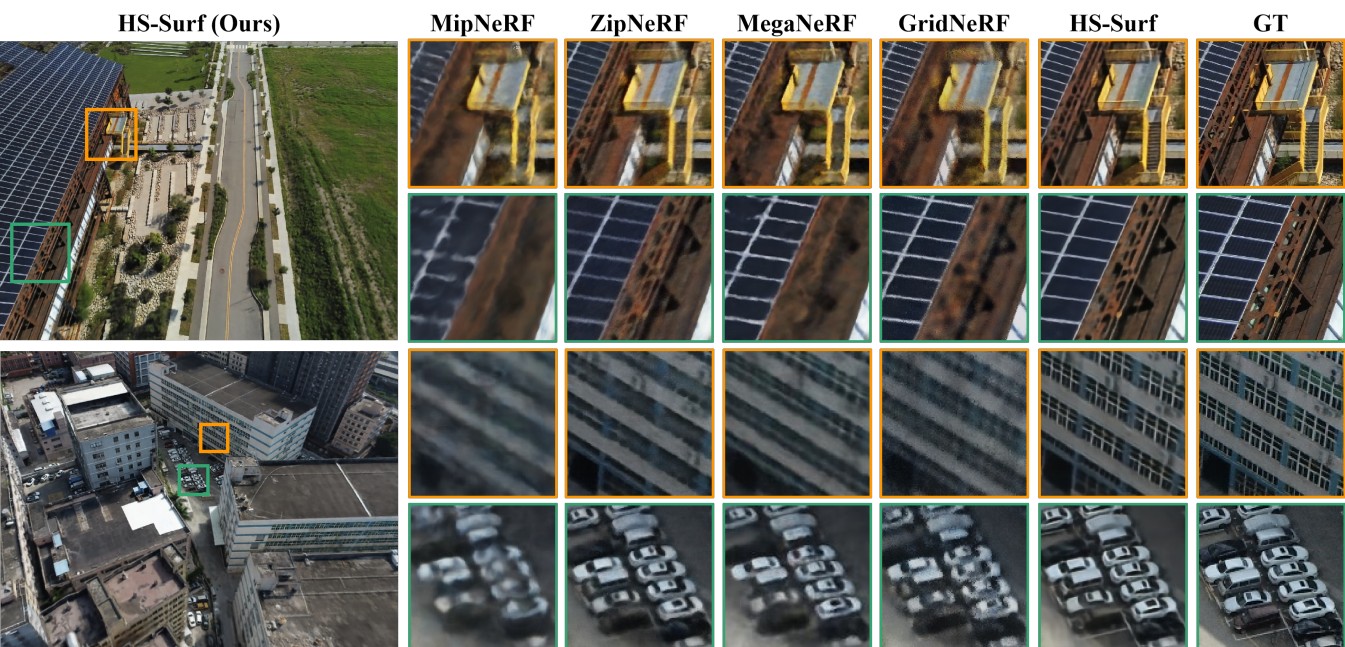

**Figure 3: A visual comparison example from the aerial orbiting photography of *Transamerica*. The top row is near the ground and the bottom row is far from the ground. The zoom-in images include different distances (near and far) and appearances (floater and buildings).**

**Figure 4: Two visual comparison examples from the drone shooting with fixed altitude and route. The top row is the *building* and the bottom row is the *residence*, respectively. The zoom-in images include the complex building structure, vertical and horizontal planes, and the vehicles with tiny height differences.**

as follows. F) The removal of feature fusion strategy in the initialization stage results in a decreased modeling ability for distance scales. In supplementary material, we demonstrate the impacts of feature fusion and compensation on rendering results. G) The removal of compensation leads to a significant decrease in model performance.

In Figure 5, the results without compensation lose a lot of high-frequency textures at the top of the building. For more results of depth and rendering, please refer to the supplementary material. H) Removing depth augmentation (dea) in compensation leads to poor geometry and holes, and it also affects rendering quality, as

**Table 3: Performance comparison of ablation experiments**

| Model | Transamerica | | | 56 Leonard | | | Building | | | Residence | | |
|---|---|---|---|---|---|---|---|---|---|---|---|---|
| | PSNR ↑ | SSIM ↑ | LPIPS ↓ | PSNR ↑ | SSIM ↑ | LPIPS ↓ | PSNR ↑ | SSIM ↑ | LPIPS ↓ | PSNR ↑ | SSIM ↑ | LPIPS ↓ |
| A) w/o fusion & PP | 23.77 | 0.7411 | 0.4087 | 24.43 | 0.7877 | 0.3476 | 20.79 | 0.4971 | 0.5060 | 21.41 | 0.5275 | 0.5394 |
| B) w/o compensate & PP | 24.95 | 0.7966 | 0.3528 | 23.24 | 0.7063 | 0.4528 | 20.50 | 0.4911 | 0.5151 | 21.68 | 0.5639 | 0.5089 |
| C) w/o dea & PP | 25.15 | 0.8089 | 0.3309 | 25.05 | 0.8262 | 0.2896 | 21.25 | 0.5521 | 0.4681 | 19.53 | 0.5010 | 0.5509 |
| D) w/o HS & PP | 25.18 | 0.8105 | 0.3332 | 25.98 | 0.8515 | 0.2613 | 20.61 | 0.5274 | 0.4852 | 21.61 | 0.5549 | 0.5106 |
| E) w/o PP | 25.25 | 0.8147 | 0.3251 | 26.12 | 0.8566 | 0.2537 | 21.52 | 0.5678 | 0.4537 | 21.99 | 0.5797 | **0.4952** |
| F) w/o fusion | 24.50 | 0.7814 | 0.3383 | 25.30 | 0.8240 | 0.2875 | 21.35 | 0.5567 | 0.4788 | 21.70 | 0.5725 | 0.5281 |
| G) w/o compensate | 25.29 | 0.8149 | 0.3146 | 23.80 | 0.7446 | 0.3722 | 20.89 | 0.5428 | 0.5011 | 21.81 | 0.5867 | 0.5128 |
| H) w/o dea | 25.49 | 0.8250 | 0.2983 | 25.56 | 0.8431 | 0.2623 | 21.66 | 0.5911 | 0.4552 | 19.99 | 0.5268 | 0.5584 |
| I) w/o HS | 25.51 | 0.8268 | 0.2988 | 26.28 | 0.8634 | 0.2419 | 21.14 | 0.5648 | 0.4711 | 21.76 | 0.5797 | 0.5185 |
| J) complete | **25.59** | **0.8304** | **0.2941** | **26.41** | **0.8679** | **0.2363** | **21.88** | **0.6039** | **0.4417** | **22.12** | **0.5982** | 0.5015 |

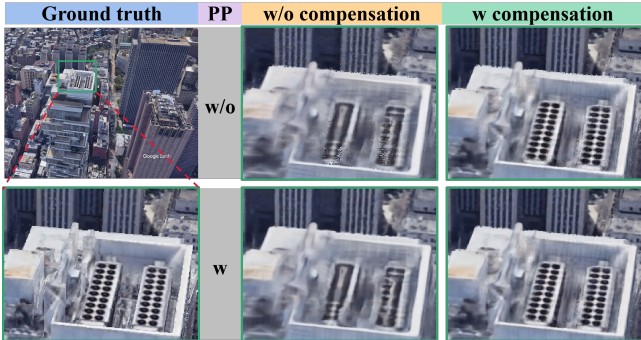

**Figure 5: Ablation study on compensation. Without compensation, the rendering results lose a lot of high-frequency textures. The post-processing (PP) just removes noises, but cannot generate the lost textures.**

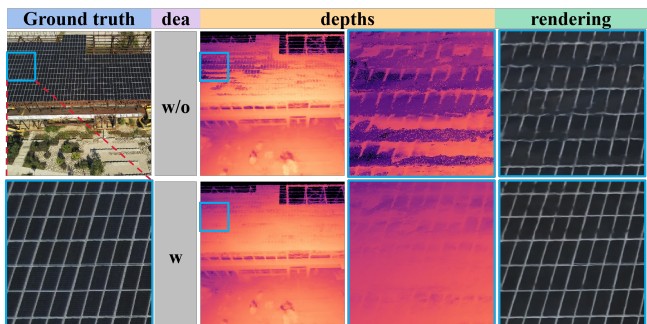

**Figure 6: Ablation study on the depth augmentation. With depth augmentation, the depth map is further improved, alleviating the holes and depth texture-copy.**

shown in Figure 6. I) Replacing the high-frequency shell with a single spatial point in compensation results in the framework's inability to model distance scales, as shown in Figure 7. A)-E) remove the post-processing based on F)-J). Therefore, the rendering results contain noise and have relatively low quality, as shown in Table 3 and Figure 5.

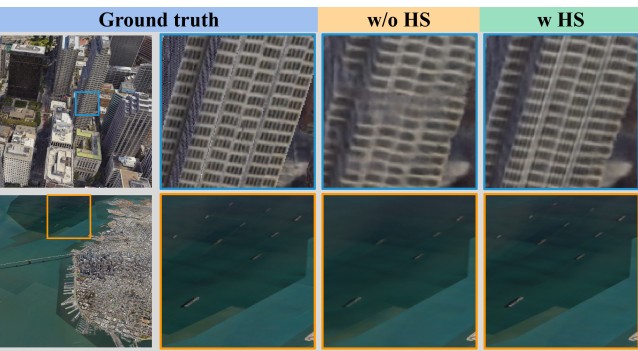

**Figure 7: Ablation study on the HS. When removing the HS and directly sampling a point on the surface, the targets like windows and ships-like become unclear.**

## 5 DISCUSSION AND CONCLUSION

In this work, we aim to improve the quality of high-frequency textures in urban and aerial large-scale scenes by dealing with issues of inefficient sampling and various distances. We have presented HS-Surf, a novel high-frequency surface shell radiance field method to improve large-scale scene rendering. We create a high-frequency shell on the scene surface under the current view, and sample conic frustums on this shell to overcome the sampling inefficiency in previous methods. As a result, model capacity is efficiently utilized to render high-frequency textures. Additionally, to model the distances with drastic changes in large-scale scenes, we embed frustums representing distance into voxel grids to construct the scene at different distance scales.

Our HS-Surf achieves better rendering results in large-scale scenes, particularly concerning high-frequency textures. The solid ablation study experiments validate the effectiveness of each component in our model. Meanwhile, as we optimize the sampling and make better use of the model capacity, our implementation speed is also faster. Furthermore, our method can easily generalize to NeRFs variants and new rendering techniques. In the future, we will explore integrating the high-frequency surface shell into other rendering techniques, and consider rendering texture details at different distance scales in dynamic scenes.

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
