# OpenReview forum: "HS-Surf: A Novel High-Frequency Surface Shell Radiance Field to Improve Large-Scale Scene Rendering"
_acmmm.org/ACMMM/2024/Conference — MM2024 Poster_

### Official Review · Reviewer_U1UZ · 2024-05-08

**Rating:** 3
**Confidence:** 3

**Summary:**

This paper proposes a method called HS-Surf for the task of large-scale scene reconstruction. It first uses a hybrid representation from instant-NGP and Mip-NeRF to build a coarse scene. Then refines the surface depth and appearance details by training two additional MLPs. Substantial experiments and ablation experiments are conducted.

**Strengths:**

- According to the reported results, the novel idea of building compensation modules is effective in refining high-frequency details for large-scene reconstruction.
- Substantial experiments and ablation experiments are conducted, with extensive quantitative and qualitative results reported.

**Limitations:**

- **Concept.** The method is called a "Surface Shell Radiance Field" that "constructs high-frequency shells on the scene surface", which seems to be hugely different from previous NeRF-based methods in representation. But actually, it heavily relies on a traditional neural field to shrink the sampling range and offer a basic appearance, and then train new NeRFs for compensation. I do not think this representation is significantly different in technique from GridNeRF [7] and instant-ngp [27] (similarly, it uses an occupancy grid to limit the sampling range, which the authors have not discussed), as the "surface shell" does not dominate the representation but is just for compensation. Meanwhile, the proposed sampling strategy is also somehow similar to some existing methods that only sample points near the surface. Overall, it's unsuitable and unnecessary to raise such a new and confusing concept to describe the proposed method. Please consider simplifying it.

- **Contribution.** The second contribution declared in this paper is to fuse the features from conical frustum with grid features at the initialization stage. However, it seems to not contain a special design for large-scale scenarios. While this part shares a similar idea with ZipNeRF [8], the authors didn't provide discussions about the differences between them. Meanwhile, as this point can not provide sufficient novel insights, it is likely not worth being individually listed in the introduction.

- **Problems in evaluation.** In Table 1,  the reported performances of the baselines, especially GridNeRF, drop tremendously from its original papers (e.g. for GridNeRF, 25.505->20.00 in PSNR of *Campus*). And I couldn't find any explanation for this in the paper. The results are less convincing in my view. More explanations are necessary about it.

- **Doubtful declaration.** In the abstract, the authors claimed existing problems are "due to insufficient model capacity on the scene surface. This is attributed to their sampling locations or grid vertices falling in empty areas". This causal relationship needs to be proven or explained. Also, the proposed method has not been verified to exceed other methods in avoiding sampling empty areas.

- **Missing work.** The paper has ignored a famous related work Switch-NeRF. Nevertheless, it may not influence the conclusions.

- **Descriptions.** 1) Are 1/50 and 1/200 generative for all scenes? Otherwise, I think the specific value should be mentioned only in the implementation details. 2) If the "Hash-based voxel grid" in Section 3 refers to instant-ngp or some other works, it should be clearly clarified by a citation.

Will consider raising the rating if the first four concerns can be well explained.

**Suitability:**

2

---

### Official Review · Reviewer_edyY · 2024-05-24

**Rating:** 5
**Confidence:** 3

**Summary:**

This research aims to improve the rendering quality of existing NeRF methods in large-scale scenes.
First, it retains high-frequency textures and thus proposes a high-frequency surface shell created based on depth information and samples it on the shell to improve the capacity utilization of the model in the surface area.
The second is the rationality of sampling, so a method of combining voxel grid and MLP features is proposed to combine the advantages of the two.
Finally, the CNN was used for post-processing to ensure the high quality of the result

**Strengths:**

Whether it is visual quality or indicator quality, this method has certain advantages over previous methods. At the same time, the surface shell sampling method proposed by this method is very novel.

**Limitations:**

Although this method can achieve very good results, its model parameters are too large, and the overall architecture is very complex compared to other methods.

**Suitability:**

3

---

### Official Review · Reviewer_eX95 · 2024-05-24

**Rating:** 4
**Confidence:** 4

**Summary:**

This paper presents a hybrid radiance field representation that integrates a high-frequency surface shell with volume density for large-scale scene reconstruction and rendering. The proposed high-frequency surface shell has an adaptive width proportional to depth, enabling detailed levels in rendered images. It is also more efficient in sampling compared to the standard volume rendering method for large-scale scene rendering. Experiments indicate that the proposed method achieves superior rendering quality compared to existing methods on large-scale scenes. An extensive ablation study is included to analyze the effectiveness of each module.

**Strengths:**

The proposed adaptive width of the surface shell is an effective approach for achieving various levels of detail in rendering, particularly beneficial for the large-scale scenes examined in this paper. It allows for rough sampling near the surface to provide fewer high-frequency details when depth is large, and more focused sampling for details when depth is small.

The rendering quality improvement of the proposed method is substantial compared to existing methods based on NeRF. The visual comparisons in Figures 3 and 4 further demonstrate the superiority of the proposed method in rendering quality.

Additionally, the paper is well-organized and clearly written, making it easy to follow the structure and understand the content presented.

**Limitations:**

The functions of integrated positional encoding (IPE) and the proposed adaptive width of the surface shell are similar  (depth increases lead to a larger frustum in IPE for less details). Since the paper employs both approaches to model different levels of rendering details, it is unclear which method contributes more to the final rendering quality.

Furthermore, the concept of the surface shell is similar to the adaptive shell in [R1]. The authors should include this reference and discuss the differences to the proposed method.

Additionally, the method of VastGaussian [R2], which addresses the same problem of large-scale scene reconstruction, should be mentioned and discussed, even though its underlying representation, based on 3D Gaussian Splatting, differs from the volume density representation in this paper.

[R1] Wang, Z., Shen, T., Nimier-David, M., Sharp, N., Gao, J., Keller, A., ... & Gojcic, Z. (2023). Adaptive Shells for Efficient Neural Radiance Field Rendering. ACM Transactions on Graphics (TOG), 42(6), 1-15.

[R2] Lin, J., Li, Z., Tang, X., Liu, J., Liu, S., Liu, J., ... & Yang, W. (2024). VastGaussian: Vast 3D Gaussians for Large Scene Reconstruction. arXiv preprint arXiv:2402.17427.

**Suitability:**

3

---

### Official Review · Reviewer_G9FK · 2024-05-25

**Rating:** 5
**Confidence:** 3

**Summary:**

The paper introduces a method aimed at enhancing the rendering quality of high-frequency textures in large-scale urban and aerial scenes. The method leverages depth-guided sampling to construct a high-frequency surface shell (HS-Surf) that envelops the scene surface, thereby improving the model's capacity to capture detailed textures.
The introduction of a high-frequency surface shell to focus model capacity on texture-rich areas is a significant contribution. This addresses the problem of inefficient sampling in empty spaces seen in prior methods.

**Strengths:**

The introduction of a high-frequency surface shell to focus model capacity on texture-rich areas is a significant contribution. The use of depth information to guide the construction of the surface shell is well-motivated and effectively reduces the sampling in empty spaces. This addresses the problem of inefficient sampling in empty spaces seen in prior methods. Then it includes substantial experiments and ablation studies demonstrating the effectiveness of HS-Surf over existing techniques. The proposed method not only improves rendering quality but also increases rendering speed by 2× to 4×, which is a notable improvement in computational efficiency.

**Limitations:**

In my opinion, this article proposes more detailed scene solutions while ensuring that rendering speed does not decrease, which is a good work. However, the paper mainly focuses on urban and aviation scenarios, but the generalization ability of this method may be challenged in more scenarios. The method used in this article seems to be a universal solution. Can it be applied in general scenarios?

**Suitability:**

3

---

### Meta-Review · Area_Chair_A7K4 · 2024-06-27

**Recommendation:** Accept (Poster)
**Confidence:** 5

**Metareview:**

This paper proposes a high-frequency surface shell to focus model capacity on texture-rich areas, which is interesting and effective. All the reviewers agree to accept this paper. The authors should carefully address the concerns in the final version, such as the clear explanations of the working principle, experimental evaluations and discussions of relevant works.